# TopoMHC: Sequence–Topology Fusion for MHC Binding

## Abstract

Accurate prediction of peptide immunogenicity, particularly the binding affinity to major histocompatibility complex (MHC) molecules, is critical for vaccine design and immunotherapy. Existing approaches are predominantly sequence-based and often overlook structural variability and topological organization, which restricts predictive reliability. In this work, we introduce a multi-modal framework that integrates sequence embeddings from a pre-trained protein language model (e.g., ESM-C) with topology-informed descriptors derived from peptide conformations. We generate peptide conformers using molecular dynamics simulations and RDKit-based methods, and from these conformations we compute persistent homology invariants, Betti numbers, geometric statistics, and residue connectivity measures. These topological features are then fused with sequence embeddings through a cross-attention mechanism, allowing the model to capture both local sequence patterns and global conformational organization. Extensive experiments demonstrate consistent improvements over conventional structure-based and sequence-only baselines, establishing state-of-the-art performance in peptide immunogenicity prediction.

## 1 Introduction

Predicting peptide immunogenicity, especially their binding to major histocompatibility complex (MHC) molecules, is central to computational immunology with applications in vaccine design and immunotherapy. Presentation of peptide fragments by MHC class I molecules is essential for T-cell receptor recognition, making accurate identification of immunogenic peptides critical for epitope discovery and cancer vaccines. Experimental assays such as ELISPOT and tetramer staining are reliable but costly and low-throughput (Schapira et al., 1999; Genheden & Ryde, 2015), motivating computational approaches.

Early models such as NetMHC employed neural networks trained on curated binding datasets and became widely used benchmarks. However, these sequence-based approaches generalize poorly across alleles and peptide lengths, often missing structural determinants of immunogenicity (Lundegaard et al., 2008a; Wang et al., 2017). Recent advances in protein representation learning, notably ESM and ProtTrans, generate contextual embeddings capturing evolutionary and biochemical information (Rives et al., 2021; Elnaggar et al., 2021), achieving strong results in structural and functional prediction. Yet, sequence-only embeddings lack explicit spatial awareness necessary for MHC recognition (Lin et al., 2022).

To address this, structure-informed learning has emerged. Graph neural networks and 3D convolutional models capture residue neighborhoods and interface geometry, improving binding prediction and protein interactions (Gainza et al., 2020; Ingraham et al., 2019). Still, they rely on coordinate-level data and overlook transformation-invariant features (Cai et al., 2024). Topological data analysis offers a complementary perspective: persistent homology encodes structural invariants across scales, robust to noise, and has shown promise in protein folding and ligand binding (Cang & Wei, 2018; Pun et al., 2018). Building on this, we propose **TopoMHC**, a multi-modal framework integrating pretrained sequence embeddings with topology-informed descriptors via bidirectional cross-attention, jointly capturing sequence motifs and structural organization for immunogenicity prediction.

**Our key contributions are:**

- A unified architecture that integrates pretrained sequence embeddings with geometric and topological representations of peptide–MHC interfaces.

- A 100-dimensional topological feature vector encompassing contact statistics, interface geometry, distance-based descriptors, and persistent homology.

- An adaptive feature fusion mechanism with cross-attention that improves generalization, validated on curated binding affinity datasets against strong baselines.

## 2 RELATED WORK

### 2.1 PHYSICS-BASED METHODS

Physics-based simulations have long been regarded as the gold standard for quantifying molecular interactions. Methods such as molecular dynamics (MD) and free energy perturbation (FEP) offer atomistic resolution of peptide–MHC interactions, while post-processing approaches such as MM/PBSA and MM/GBSA approximate binding free energy through energy decomposition (Genheden & Ryde, 2015; Wang et al., 2019). Although accurate in principle, these simulations are computationally demanding, require extensive conformational sampling, and are sensitive to force-field parametrization. Such constraints make them unsuitable for large-scale immunogenic peptide screening or clinical applications where thousands of candidates must be evaluated.

### 2.2 MACHINE LEARNING APPROACHES

To improve scalability, a wide range of machine learning and deep learning models have been developed for MHC-related tasks. Classical approaches such as COMBINE analysis applied regression over energy decomposition profiles to approximate binding affinity (Ganotra & Wade, 2018), while scoring-based models including Syfpeithi (Rammensee et al., 1999) and PickPocket (Lundegaard et al., 2008b) relied on motif libraries or position-specific weight matrices. Although these methods offered early breakthroughs, their reliance on predefined descriptors or allele-specific motifs severely limited generalization across diverse peptide repertoires. With the availability of large-scale immunological resources such as IEDB (Vita et al., 2019), neural models became dominant, including frameworks like NetMHCpan (Jurtz et al., 2017; Reynisson et al., 2020), MHCflurry (O'Donnell et al., 2018), and DeepLigand (Han et al., 2023), which employed artificial neural networks, convolutional encoders, or recurrent architectures trained on binding affinity and eluted ligand data. More recently, geometric and structural cues have been leveraged: graph neural networks and 3D deep learning capture residue-level contacts and interfacial geometry (Gligorijevic et al., 2021; Townshend et al., 2021), while GearBind (Cai et al., 2024), DeepPPAPred (Chakrabarty et al., 2025), and FuncPhos-STR (Zhang et al., 2024) integrate graph-based transformers, 3D convolutions, or AlphaFold-derived dynamics. Despite these advances, existing methods remain biased toward explicit coordinate-level features and often overlook global invariants that remain stable across conformational variations.

### 2.3 TOPOLOGICAL DATA ANALYSIS

Topological data analysis provides an orthogonal perspective by quantifying structural invariants that are robust to deformation. Persistent homology has been successfully applied to protein folding, docking, and ligand binding, offering stable multiscale descriptors of shape and connectivity (Cang & Wei, 2018; Hofer et al., 2017; Pun et al., 2018). Recent studies further suggest that topology captures organizing principles of the protein universe, such as conserved folds and functional domains, which are invisible to purely geometric representations (Madsen et al., 2025). Nevertheless, the integration of topology into peptide–MHC modeling remains limited (Liu et al., 2022), leaving an open opportunity for multi-modal approaches that combine sequence embeddings, structural geometry, and topology to improve predictive robustness. Our work situates itself at this intersection, extending beyond conventional sequence-based or structure-only methods to leverage topological descriptors for immunogenicity prediction.

## 3 METHOD

We propose **TopoMHC**, a multi-modal deep learning framework that integrates protein sequence embeddings with topology-informed structural descriptors to predict peptide immunogenicity.

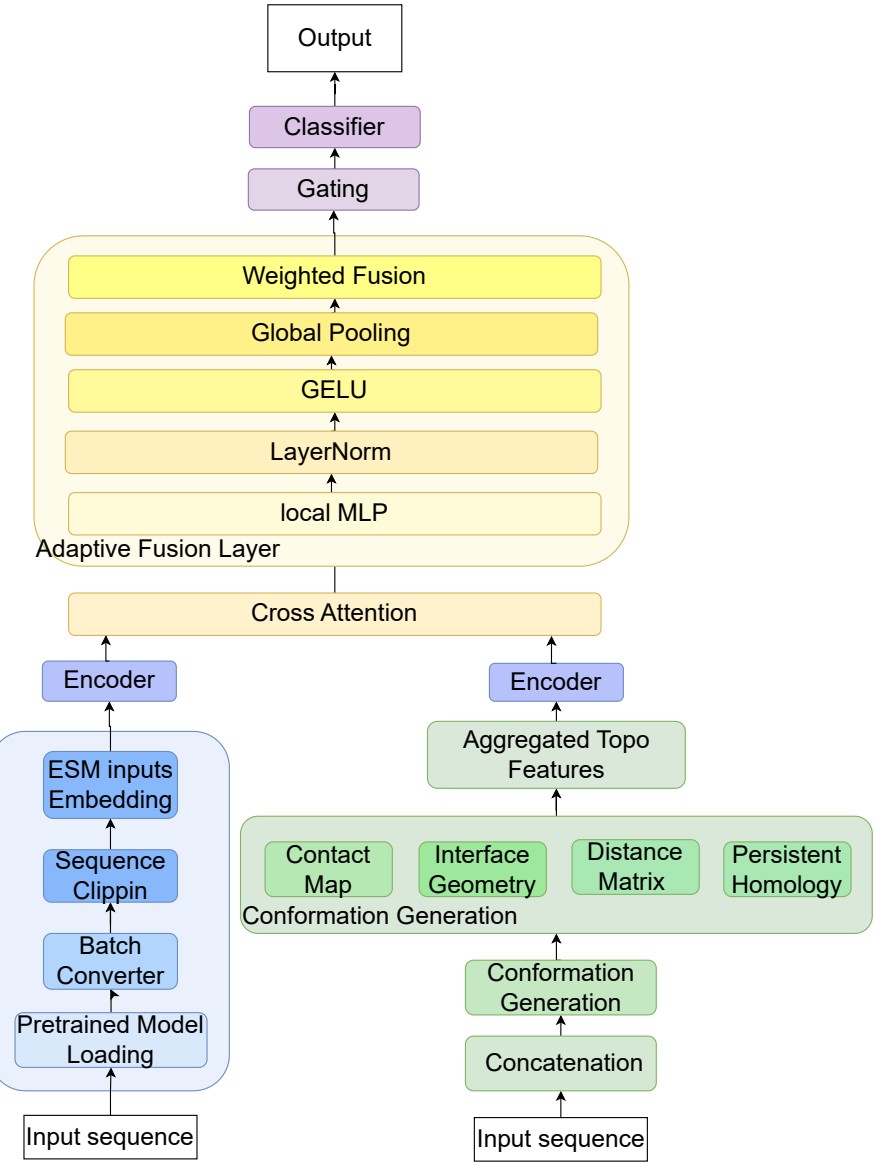

Figure 1: Overall architecture of the proposed **TopoMHC** framework. Peptide sequences are first encoded into token-level embeddings using a pretrained protein language model (i.e., ESMC), which are summarized into a sequence representation. In parallel, 3D conformations are generated with RDKit and used to compute topological descriptors, including contact maps, interface geometry, distance statistics, and persistent homology features. The two modalities are projected into a shared latent space and aligned through bi-directional cross-attention. An AFF module together with an explicit gating mechanism integrates the sequence- and topology-derived representations. The fused embedding, concatenated with an independent topology encoder, is passed to a gated residual classifier for final immunogenicity prediction.

### 3.1 PROBLEM FORMULATION

Let each peptide be represented by an amino acid sequence $S = (s_1, s_2, \ldots, s_L)$ of length $L$, together with its experimentally determined immunogenicity label $y \in \{0, 1\}$. Our goal is to learn a mapping

$$f_\theta : \mathbb{R}^{d_s \times L} \times \mathbb{R}^{d_t} \longrightarrow [0, 1],$$

where $\mathbb{R}^{d_s \times L}$ denotes the token-level sequence embeddings extracted from a pretrained protein language model (ESM), and $\mathbb{R}^{d_t}$ denotes the handcrafted topological feature vector derived from 3D conformations. The output $f_\theta(x)$ corresponds to the probability that peptide $S$ is immunogenic.

Given a training set $\mathcal{D} = \{(S^{(i)}, x_{\text{seq}}^{(i)}, x_{\text{topo}}^{(i)}, y^{(i)})\}_{i=1}^N$, the learning objective is to minimize the empirical cross-entropy loss:

$$\mathcal{L}(\theta) = -\frac{1}{N} \sum_{i=1}^N \left[ y^{(i)} \log f_\theta(x^{(i)}) + (1 - y^{(i)}) \log \left(1 - f_\theta(x^{(i)})\right) \right].$$

where $x^{(i)} = (x_{\text{seq}}^{(i)}, x_{\text{topo}}^{(i)})$ is the concatenation of sequence and topology features.

At inference time, the model outputs $\hat{y} = f_\theta(x)$, which is thresholded at $0.5$ to determine the predicted immunogenicity class.

### 3.2 SEQUENCE REPRESENTATION

We concatenate the antibody and antigen sequences as $S = [S_a; S_g]$ and encode them with a pretrained protein language model from the ESM family (Rives et al., 2021; Elnaggar et al., 2021). This produces residue-level embeddings

$$H \in \mathbb{R}^{L \times d},$$

where $L$ is the combined sequence length and $d$ the embedding dimension. A mean pooling operation summarizes $H$ into a global descriptor,

$$x_{\text{seq}} = \frac{1}{L} \sum_{i=1}^L H_i,$$

which is then passed through a two-layer MLP with LayerNorm, GELU, and Dropout to yield the sequence representation

$$h_{\text{seq}} \in \mathbb{R}^{d_h}.$$

### 3.3 CONFORMATION GENERATION

For each peptide sequence $S = (s_1, \ldots, s_L)$, we construct 3D conformations using RDKit with ETKDG initialization and UFF optimization. Formally, a conformation is represented as a set of atomic coordinates

$$X = \{x_i \in \mathbb{R}^3\}_{i=1}^n,$$

where $n$ is the number of atoms. To account for conformational variability, we generate $N_c$ conformers per peptide (default $N_c = 10$), yielding

$$\mathcal{C}(S) = \{X^{(1)}, X^{(2)}, \ldots, X^{(N_c)}\}.$$

If RDKit fails to embed a valid structure, a fallback procedure generates simplified backbone-based coordinates with stochastic perturbations, ensuring each peptide has at least $N_c$ candidate conformations.

### 3.4 TOPOLOGICAL FEATURE CONSTRUCTION

We extract $x_{\text{topo}}$ from four geometric representations of the antibody-antigen structure. These features capture multiscale interaction patterns from contact connectivity, interfacial geometry, Euclidean distances, and topological persistence.

**Contact Map Features.** We construct a binary matrix $C \in \{0,1\}^{n \times n}$ indicating whether a pair of residues is in contact (e.g., $C_{ij} = 1$ if the $C_\alpha$-$C_\alpha$ distance is below threshold $\delta$). Based on this, we define:

$$D = \frac{1}{n^2} \sum_{i,j} C_{ij}$$

*Contact density*, measuring the overall sparsity of the contact network.

$$k_i = \sum_j C_{ij}, \quad \mu_k = \frac{1}{n} \sum_i k_i$$

$k_i$ is the *contact degree* of residue $i$, and $\mu_k$ is the average number of contacts per residue.

$$\sigma_k = \sqrt{\frac{1}{n} \sum_i (k_i - \mu_k)^2}, \quad k_{\max} = \max_i k_i$$

Standard deviation $\sigma_k$ quantifies structural irregularity, and $k_{\max}$ identifies hotspots with maximal connectivity.

$$\bar{C} = \frac{\mathrm{Tr}(C^3)}{\sum_i k_i(k_i - 1)}$$

*Clustering coefficient* $\bar{C}$ reflects the local density of triangles in the graph, indicating compactness and modularity.

**Interface Geometry Features.** For two chains (antibody and antigen), we define the binary interface contact matrix $I \in \{0,1\}^{n_a \times n_g}$. From this we compute:

$$D_I = \frac{1}{n_a n_g} \sum_{i=1}^{n_a} \sum_{j=1}^{n_g} I_{ij}$$

*Interface density*, capturing how tightly the two proteins interact.

$$R_a = \frac{1}{n_a} \sum_{i=1}^{n_a} \mathbb{1}\!\left( \sum_{j=1}^{n_g} I_{ij} > 0 \right), \quad R_g = \frac{1}{n_g} \sum_{j=1}^{n_g} \mathbb{1}\!\left( \sum_{i=1}^{n_a} I_{ij} > 0 \right)$$

*Interface coverage*, measuring the proportion of residues involved in cross-chain contacts.

$$\mu_a = \frac{\sum_{i=1}^{n_a} a_i \mathbb{1}(a_i > 0)}{\sum_{i=1}^{n_a} \mathbb{1}(a_i > 0)}, \quad \mu_g = \frac{\sum_{j=1}^{n_g} g_j \mathbb{1}(g_j > 0)}{\sum_{j=1}^{n_g} \mathbb{1}(g_j > 0)}$$

*Mean interface degree*, assessing how many contacts each interacting residue participates in.

**Distance Map Features.** From the Euclidean distance matrix $D \in \mathbb{R}^{n \times n}$ between residues, we compute:

$$d_{\min} = \min_{i,j} D_{ij}, \quad \mu_D = \frac{1}{n^2} \sum_{i,j} D_{ij}$$

*Minimum* and *mean internal distance* measure spatial compactness.

$$d_{\mathrm{med}} = \mathrm{median}(\{D_{ij}\})$$

A robust estimator of internal packing.

$$\mu_I = \frac{1}{|D_I|} \sum_{(i,j) \in D_I} D_{ij}, \quad N_c = |\{(i,j) : D_{ij} \leq \delta\}|$$

*Interface mean distance* $\mu_I$ and *contact count* $N_c$ summarize interaction range and density.

**Persistent Homology Features.** Persistent homology (PH) is a key method in topological data analysis (TDA), designed to quantify multiscale topological structures in geometric data. In our setting, PH captures the evolution of connected components, loops, and voids formed by atoms in 3D space, offering descriptors that are invariant to spatial transformations and robust to noise.

Given a point cloud $X = \{x_i \in \mathbb{R}^3\}_{i=1}^n$ extracted from atomic coordinates (e.g., heavy atoms or $C_\alpha$), we construct a Vietoris–Rips filtration: for a threshold $\epsilon > 0$, a simplex is added for each group of points with pairwise distance less than $\epsilon$. As $\epsilon$ increases, simplicial complexes $K_\epsilon$ are formed, and topological features appear (birth) and vanish (death).

Each topological feature corresponds to a persistence pair $(b_i^k, d_i^k)$ in dimension $k$. From these, we compute lifetimes $\ell_i = d_i^k - b_i^k$ and derive the following descriptors:

$$N_k = |\{(b_i^k, d_i^k)\}|, \quad \mu_k = \frac{1}{N_k} \sum_i \ell_i,$$

$$\sigma_k = \sqrt{\frac{1}{N_k} \sum_i (\ell_i - \mu_k)^2}, \quad \ell_{\max} = \max_i \ell_i$$

### 3.5 CROSS-MODAL FUSION

To integrate the sequence representation $h_{\text{seq}}$ and the topology representation $\tilde{x}_{\text{topo}}$, we design a cross-modal fusion module composed of bi-directional cross-attention, adaptive feature fusion (AFF), and an explicit gating mechanism.

**Bi-directional cross-attention.** Both modalities are first projected into a shared hidden space of dimension $d_h$:

$$t = W_t \tilde{x}_{\text{topo}} \in \mathbb{R}^{d_h}, \qquad S = W_s H \in \mathbb{R}^{L \times d_h},$$

where $H$ are token-level ESM embeddings and $W_t, W_s$ are learnable projections. We expand $t$ into $T$ pseudo-tokens $\{t_1, \ldots, t_T\}$ ($T{=}1$ by default) to serve as queries. Cross-attention is then performed in two directions:

$$H_{t \to s} = \text{MHA}(Q = T, K = S, V = S) \in \mathbb{R}^{T \times d_h},$$

$$H_{s \to t} = \text{MHA}(Q = S, K = T, V = T) \in \mathbb{R}^{L \times d_h},$$

where MHA denotes multi-head attention. We aggregate the outputs by averaging across tokens:

$$h_{t \to s} = \frac{1}{T} \sum_{j=1}^T H_{t \to s, j}, \qquad h_{s \to t} = \frac{1}{L} \sum_{i=1}^L H_{s \to t, i}.$$

**Adaptive feature fusion (AFF).** To combine the two directional summaries, we employ an adaptive weighting module that predicts fusion weights $(\alpha, \beta)$:

$$[\alpha, \beta] = \text{Softmax}(W_f [h_{t \to s}; h_{s \to t}]),$$

yielding the fused representation

$$h_{\text{bi}} = \alpha \, h_{t \to s} + \beta \, h_{s \to t}.$$

**Explicit gating.** Finally, we stabilize the fusion by mixing $h_{\text{bi}}$ with the original topology projection $t$ using a learnable gate:

$$g = \sigma(W_g [h_{\text{bi}}; t] + b_g),$$

$$h_{\text{fuse}} = g \cdot h_{\text{bi}} + (1 - g) \cdot t,$$

where $\sigma(\cdot)$ is the sigmoid function and $h_{\text{fuse}} \in \mathbb{R}^{d_h}$ is the final cross-modal embedding passed to the prediction head.

## 3.6 PREDICTION HEAD

The fused cross-modal representation $h_{\text{fuse}} \in \mathbb{R}^{d_h}$ is concatenated with the independently encoded topology vector $h_{\text{topo}} \in \mathbb{R}^{d_h/2}$ to form the final feature:

$$z = [\, h_{\text{fuse}};\ h_{\text{topo}} \,] \in \mathbb{R}^{\frac{3}{2} d_h}.$$

**Classifier architecture.**   To stabilize training and enhance nonlinearity, we employ a residual classifier built on gated residual networks (GRN) and gated GELU units (GEGLU):

$$h_1 = \text{GRN}(\text{LayerNorm}(z)),$$
$$h_2 = \text{Dropout}(h_1),$$
$$h_3 = \text{GEGLU}(h_2),$$
$$\hat{y} = \text{Softmax}(W_o h_3 + b_o),$$

where $\hat{y} \in [0,1]^2$ denotes the predicted probability distribution over immunogenic and non-immunogenic classes.

**Training objective.**   We optimize the standard cross-entropy loss:

$$\mathcal{L}(\theta) = -\frac{1}{N} \sum_{i=1}^{N} \Big( y^{(i)} \log \hat{y}_1^{(i)} + (1 - y^{(i)}) \log \hat{y}_0^{(i)} \Big),$$

where $\hat{y}_1^{(i)}$ and $\hat{y}_0^{(i)}$ are the predicted probabilities for positive and negative classes, respectively.

At inference time, we output $\hat{y} = \arg\max_j \hat{y}_j$ as the predicted label, with $\hat{p} = \hat{y}_1$ serving as the immunogenicity confidence score.

## 4 EXPERIMENTS

We conduct experiments to systematically evaluate the performance of **TopoMHC** on peptide–MHC immunogenicity prediction. The primary goal is to examine whether incorporating topology-informed structural descriptors provides consistent benefits over sequence-only baselines. To this end, we perform direct baseline comparisons under identical training and evaluation protocols. All experiments are carried out on a curated binding affinity dataset, and results are reported using standard classification metrics to ensure reproducibility and fair comparison.

### 4.1 DATASET

We evaluate our framework on a curated binary classification dataset of peptide–MHC binding affinity (BA). Specifically, we focus on the HLA-A*02:01 allele, one of the most extensively studied MHC class I molecules due to its high prevalence and clinical relevance. The dataset contains **12,953** peptide samples, each paired with an immunogenicity label that indicates whether the peptide is experimentally validated as a binder or a non-binder. To ensure consistency, all sequences are retained within the canonical length range of 8–11 amino acids, which matches the binding preference of class I MHC molecules. This dataset provides a reliable benchmark for evaluating large-scale immunogenicity prediction models and has been widely used in the development of existing baseline methods.

### 4.2 EXPERIMENTAL SETUP

We formulate peptide–MHC binding prediction as a binary classification task. Following standard practice (Lundegaard et al., 2008a), peptides are labeled as positive (binder) if their measured or predicted IC50 is below 500 nM, and negative otherwise. This threshold is widely used in immunogenicity prediction, ensuring comparability with existing methods. The curated dataset is stratified into training, validation, and test sets with ratios of 80%/10%/10%. Stratification preserves the positive-to-negative class ratio in each split and prevents data leakage between sets. Performance is evaluated with Accuracy, Precision, Recall, F1-score, and AUC, which together provide a comprehensive view of discriminative ability and balance between false positives and false negatives. All experiments are repeated with five random seeds and results are averaged.

**Training Details.** All models are trained with AdamW (learning rate $3 \times 10^{-4}$, weight decay $1 \times 10^{-4}$) and batch size 32. We apply ReduceLROnPlateau with patience 7 and early stopping with patience 15, using validation AUC as the criterion. All experiments are conducted on a single NVIDIA A100 GPU with mixed-precision training.

## 4.3 BASELINE COMPARISON

To contextualize our results, we compare **TopoMHC** against several widely adopted baselines for peptide–MHC class I binding prediction. These methods are considered state-of-the-art in immunoinformatics and serve as standard references in epitope prediction tasks.

| Method | Acc ↑ | Prec ↑ | Rec ↑ | F1 ↑ | AUC ↑ |
|---|---|---|---|---|---|
| **TopoMHC** | **0.8510** | **0.8515** | **0.8510** | **0.8512** | **0.9081** |
| NetMHC4.0 | 0.8210 | 0.8621 | 0.6458 | 0.7384 | 0.8798 |
| NetMHCpan-4.2 | 0.8289 | 0.8381 | 0.6791 | 0.7503 | 0.9056 |
| NetMHCcons | 0.8213 | 0.8638 | 0.6450 | 0.7385 | 0.8772 |
| NetMHCstabpan | 0.8225 | 0.8691 | 0.6434 | 0.7394 | 0.8764 |

Table 1: Performance comparison between our proposed **TopoMHC** and representative baselines on the peptide–MHC binding affinity dataset.

Table 1 compares the performance of **TopoMHC** with representative baseline models. TopoMHC achieves the best overall results across all five metrics, reaching an accuracy of 0.8510 and an AUC of 0.9081. In contrast, sequence-based baselines such as NetMHC4.0, NetMHCcons, and NetMHC-stabpan show higher precision (around 0.86) but substantially lower recall (around 0.64), indicating that they tend to miss positive binders. NetMHCpan-4.2 performs more competitively, with balanced precision and recall and an AUC of 0.9056, yet still falls short of TopoMHC. These results confirm that incorporating topology-informed structural features enables TopoMHC to achieve more consistent and well-rounded predictive performance compared to sequence-only models.

## 4.4 ABLATION STUDIES

We next assess the contribution of individual components in **TopoMHC** through ablation experiments. We evaluate five variants: (i) w/o AFF (replacing the Attentional Feature Fusion with simple concatenation); (ii) w/o Cross-Attention (removing bidirectional cross-attention between sequence and topology); (iii) w/o Gate (removing the gating mechanism after fusion); (iv) ESM-only (sequence embeddings only); and (v) Topo-only (topology features only).

| Modules | | | | | Metrics | | | | |
|---|---|---|---|---|---|---|---|---|---|
| AFF | Cross-Attn | Gate | Seq | Topo | Acc ↑ | Prec ↑ | Rec ↑ | F1 ↑ | AUC ↑ |
| ✓ | ✓ | ✓ | ✓ | ✓ | **0.8510** | **0.8515** | **0.8510** | **0.8512** | **0.9081** |
| ✗ | ✓ | ✓ | ✓ | ✓ | 0.8387 | 0.8426 | 0.8387 | 0.8398 | 0.9039 |
| ✓ | ✗ | ✓ | ✓ | ✓ | 0.8375 | 0.8380 | 0.8375 | 0.8378 | 0.9035 |
| ✓ | ✓ | ✗ | ✓ | ✓ | 0.8286 | 0.8286 | 0.8286 | 0.8286 | 0.8955 |
| ✗ | ✗ | ✗ | ✓ | ✗ | 0.8310 | 0.8386 | 0.8310 | 0.8327 | 0.9028 |
| ✗ | ✗ | ✗ | ✗ | ✓ | 0.6692 | 0.6565 | 0.6692 | 0.6465 | 0.6820 |

Table 2: Ablation study of **TopoMHC** on the BA dataset. We investigate the effect of adaptive feature fusion, bidirectional cross-attention, and explicit gating. The best results are highlighted with a gray background, while the best and second-best values are additionally marked in **bold** and underlined. Separate "Seq only" and "Topo only" rows illustrate the contribution of each modality in isolation.

The ablation results confirm that each module contributes complementary value. Removing AFF lowers both Accuracy and AUC, demonstrating that adaptive weighting of topological subspaces improves generalization. Similarly, eliminating cross-attention reduces performance by over 1%, highlighting its role in explicit alignment of sequence and structural modalities. The gating mechanism, though lightweight, also proves beneficial; its removal decreases Accuracy and AUC by about

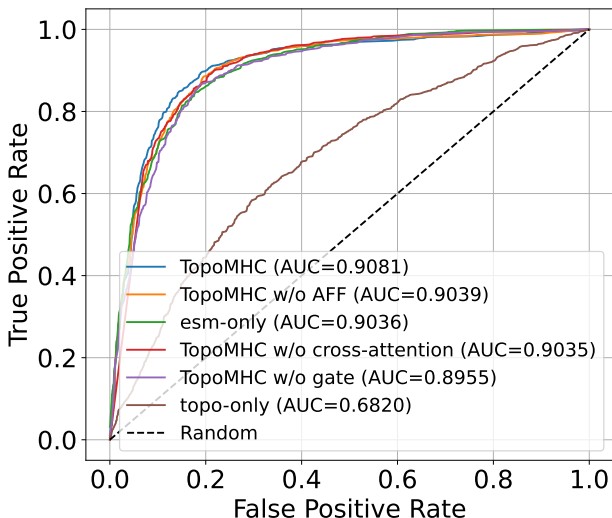

Figure 2: ROC curves for ablation variants of TopoMHC. Removing AFF or cross-attention reduces performance, while unimodal baselines (*ESM-only* and *Topo-only*) show significantly inferior results.

2%. By contrast, unimodal baselines perform markedly worse: ESM-only is competitive but lags behind multimodal variants, while Topo-only collapses to an AUC of 0.6820, underscoring that topology alone is insufficient. Together, these results show that the synergy of sequence embeddings and topology descriptors, mediated by AFF and cross-attention, is essential for the robust performance of **TopoMHC**.

## 5 CONCLUSION

In this work, we introduced **TopoMHC**, a multi-modal framework for peptide–MHC immunogenicity prediction that integrates pretrained sequence embeddings with topology-informed structural descriptors. By combining ESM-based sequence representations with handcrafted geometric and persistent homology features, our model jointly captures local sequence motifs and global structural organization. Extensive experiments on curated binding affinity datasets demonstrate that TopoMHC consistently outperforms sequence-only and traditional baselines. Ablation studies further validate the contributions of adaptive feature fusion and bidirectional cross-attention, underscoring the importance of explicit cross-modal alignment. These findings highlight the utility of incorporating topological priors into deep learning pipelines for immunogenicity prediction.

**Future work.** Several challenges remain. The dataset size is modest compared to other domains, and generalization to unseen alleles or longer peptides requires further validation. Moreover, conformer generation and persistent homology introduce additional computational cost, which may hinder large-scale deployment. Future directions include leveraging large-scale structural pretraining and developing efficient differentiable topology modules. Overall, TopoMHC provides a principled step toward unifying sequence modeling and topological analysis, opening new opportunities for computational immunology and vaccine design.

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

# A APPENDIX

## A.1 LLM USABLE STATEMENT

Large Language Models (LLMs) were only used to polish the language of this paper. No LLM was used to generate research ideas, experiments, or analyses.

## A.2 DATASET DETAILS

We conduct our experiments on a curated peptide–MHC binding affinity (BA) dataset focused on the HLA-A*02:01 allele. After preprocessing, the final dataset contains **12,953** unique peptide–HLA pairs, with **4,899** positives (immunogenic binders) and **8,054** negatives (non-binders). All peptides fall within the canonical MHC-I length range of 8–11 amino acids, consistent with class I binding preferences (Lundegaard et al., 2008a). Each example comprises a unique identifier, the raw amino acid sequence, and a binary immunogenicity label derived from experimental measurements.

**Task definition.** Following common practice in BA prediction, we formulate peptide immunogenicity as a **binary classification** problem (binder vs. non-binder). We retain only HLA-A*02:01 instances to avoid cross-allele confounding and to emphasize within-allele generalization.

**Preprocessing.** We remove duplicated entries by (*i*) exact-sequence matching and (*ii*) peptide–allele pair de-duplication. Peptides outside 8–11 residues are excluded. All labels originate from experimental assays reported in the source collections; no heuristic relabeling is applied.

**Data splits.** We adopt an **8:1:1** stratified split for training, validation, and testing, preserving the positive/negative ratio across splits. All baselines and our model are trained and evaluated under identical partitions and metrics to ensure fair comparison and reproducibility.

**Feature extraction.** Sequence embeddings are obtained from a pretrained protein language model (ESM-C), with token-level representations pooled into global descriptors. For structure-aware signals, we generate peptide conformers using RDKit (ETKDG initialization followed by UFF relaxation), from which we compute topology-informed features, including contact statistics, interface geometry descriptors, distance-based summaries, and persistent homology invariants. The resulting sequence and topology features are standardized independently and cached for efficient training and evaluation.

**Recommended diagnostics.** For transparency and reproducibility, we report: (i) class balance, (ii) peptide-length coverage within 8–11 residues, and (iii) split-wise sample counts. To aid reviewers, we also provide simple visual summaries (bar charts) of class balance and split composition, and a histogram over peptide lengths (Appendix Figures 3–5). These do not introduce new results but contextualize the evaluation setting.

## A.3 CONFORMATION GENERATION

For each peptide sequence, we generated candidate 3D conformations using RDKit-based molecular modeling pipelines. The procedure consists of four stages:

**Residue assembly.** Each amino acid in the input peptide was mapped to a canonical SMILES fragment with appropriate stereochemistry. Fragments were covalently linked in sequential order, ensuring peptide bond planarity and correct chirality at $\alpha$-carbons. The resulting molecular graph was sanitized using RDKit routines to verify valence and hydrogen counts.

**Initial embedding.** From the peptide molecular graph, we generated multiple 3D conformers using the ETKDG algorithm , which balances knowledge-based torsional preferences with distance geometry sampling. Default generation produced up to 50 conformers per peptide before filtering. To capture local flexibility, glycine residues were allowed enhanced torsional variation, while proline rings were constrained to remain planar.

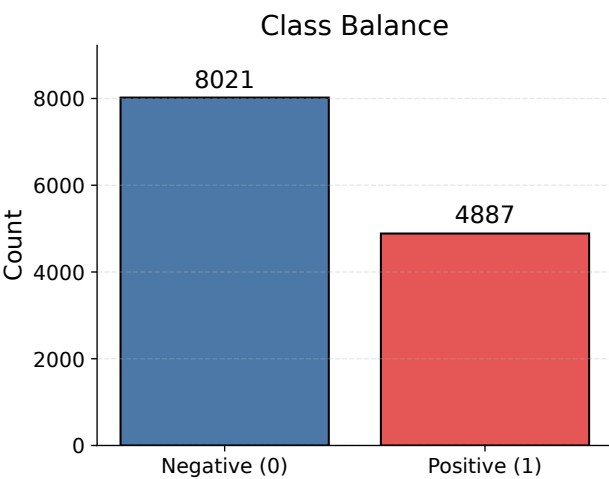

Figure 3: Class balance (positive vs. negative) in the curated HLA-A*02:01 BA dataset.

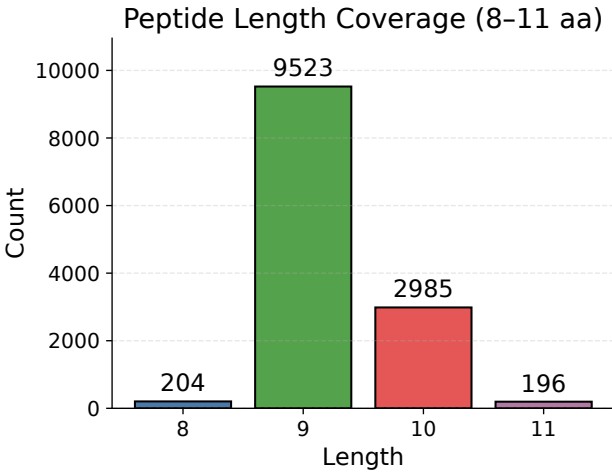

Figure 4: Peptide length coverage within 8–11 residues.

**Energy minimization and sampling.** Each raw conformer was optimized using the Universal Force Field (UFF). This removed steric clashes and improved geometric plausibility. To further explore conformational space, we applied stochastic perturbations and short molecular dynamics–like relaxation steps, producing up to $N_c = 10$ low-energy conformations per peptide. Redundant conformers were pruned by RMSD clustering with a threshold of 0.5 Å.

**Output.** For each peptide, the final conformer set was exported in pdb format. Conformations were indexed by peptide ID and conformer index, then stored for downstream feature extraction. If embedding failed, a simplified backbone-only scaffold with random torsional noise was substituted, ensuring that every peptide contributed a valid set of conformers.

This multi-stage procedure balances geometric plausibility with structural diversity, producing reproducible yet flexible peptide representations for topological feature computation.

A.4 FEATURE EXTRACTION

**Topological and geometric descriptors.** Given atomic coordinates, we computed topology-informed descriptors using `topo.py` and `topoMHC.py`. Four categories of features were extracted: (i) *contact maps* defined at multiple $C_\alpha$–$C_\alpha$ distance thresholds (default $\delta = 6$Å, 8Å), from

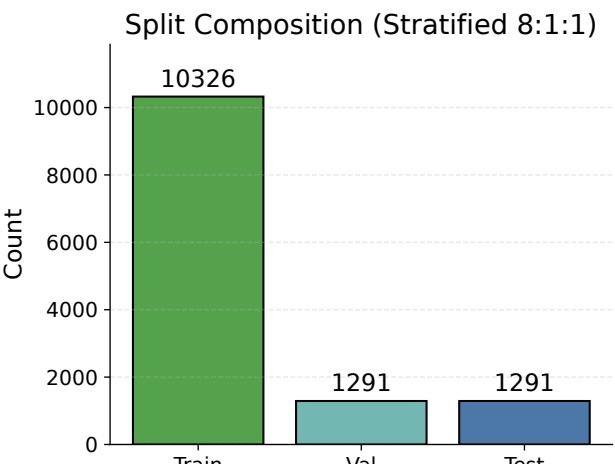

Figure 5: Train/validation/test split composition (stratified 8:1:1).

which density, degree statistics, and clustering coefficients were derived; (ii) *interface geometry* descriptors quantifying cross-chain contact coverage, mean interface degree, and residue-level participation; (iii) *distance statistics* summarizing minimum, mean, and median inter-residue distances, as well as interface mean distance and contact counts; (iv) *persistent homology* features computed from Vietoris–Rips filtrations of atomic point clouds using the gudhi backend, including Betti numbers (0–3), lifetime distributions, and summary statistics (mean, max, variance). To mitigate conformational noise, descriptors from multiple conformers were aggregated by mean, standard deviation, minimum, and maximum, producing a fixed-length feature vector per peptide.

**Sequence embeddings (ESM-C).** Sequence-level features were extracted using the ESM-C protein language model, implemented in esm_feature_extractor.py. Residue-level embeddings with hidden dimension $d = 1152$ (for the chosen ESM-C variant) were stored in float32. For efficiency, token-level embeddings were directly consumed by the fusion module without intermediate pooling. For diagnostic purposes, pooled statistics (mean, max, std) were also computed and cached. All embeddings were aligned to peptide identifiers to enable multimodal fusion with topological descriptors.

**Normalization and storage.** Prior to training, all handcrafted descriptors were z-score normalized across the dataset and serialized into .pkl dictionaries containing feature_names and feature_values. Sequence embeddings were saved as separate esm_features.pkl files, ensuring modular reuse across experiments. At runtime, features were dynamically loaded and concatenated within the data.py loader to form multimodal inputs.

MODEL ARCHITECTURE

Our classifier integrates topology and sequence modalities with a *bidirectional* cross-attention block, an Attentional Feature Fusion (AFF) layer, and an explicit gate:

- **Unified projections.** Topology features and per-token sequence embeddings are projected into a shared hidden space (dimension 256 by default).

- **Topo tokens.** A single (or few) "pseudo-topology token(s)" are formed by linearly expanding the topology vector, enabling token-to-token attention (topo_tokens=1 by default).

- **Bidirectional cross-attention.** (i) *topo → seq*: topology token(s) query sequence tokens; (ii) *seq → topo*: sequence tokens query topology token(s). Each direction uses a shared multi-head projection module (MultiHeadProj) with $H$ heads (default $H=8$) and LayerNorm+GRN feed-forward refinement.

- **AFF fusion.** The two directional summaries are fused by AFF: $\mathbf{z} = \alpha \, \mathbf{u} + \beta \, \mathbf{v}$, where $(\alpha, \beta)$ are softmax-normalized attention weights estimated from both local (LayerNorm+MLP) and lightweight global cues.

- **Explicit gate to topology prior.** A learnable sigmoid gate mixes the AFF output with the original topology encoding to stabilize training and preserve structure-aware priors.

- **Head.** We concatenate the cross-fused vector with an independent topology encoder (two-layer MLP with GELU), and feed the result into a GRN+GEGLU classifier head (LayerNorm $\rightarrow$ GRN $\rightarrow$ GEGLU $\rightarrow$ Linear) for binary prediction.

## A.5 BASELINES AND EVALUATION

**Baseline methods.** We compared **TopoMHC** against several representative peptide–MHC binding predictors.

**NetMHC4.0.** NetMHC4.0 employs fully connected neural networks trained on large-scale peptide–MHC binding affinity data from IEDB and other curated resources (Andreatta & Nielsen, 2016). It incorporates gapped sequence alignment to enhance recognition of binding motifs and is one of the most commonly used predictors for class I epitopes.

**NetMHCpan-4.2.** NetMHCpan represents a pan-specific extension of NetMHC, trained on both binding affinity and eluted ligand datasets. The latest version, NetMHCpan-4.2, integrates transfer learning and structural information to improve predictions of CD8+ T-cell epitopes across multiple HLA alleles (Reynisson et al., 2025).

**NetMHCstabpan.** Unlike predictors that only estimate binding affinity, NetMHCstabpan focuses on peptide–MHC complex stability, which has been shown to be a strong correlate of immunogenicity. It uses artificial neural networks trained on stability data across multiple alleles (Rasmussen et al., 2016).

**NetMHCcons.** NetMHCcons is a consensus-based predictor that combines the outputs of NetMHC, NetMHCpan, and PickPocket into a unified prediction score, thereby improving robustness and accuracy across alleles (Karosiene et al., 2012). This approach demonstrates the value of ensemble learning in peptide–MHC binding prediction.

**Input processing.** All baseline methods were provided with the same curated BA dataset (HLA-A*02:01, 12,953 samples). Peptides were represented by raw amino acid sequences in FASTA or CSV format, depending on the interface requirements of each model. No structural features were used in baseline runs. For reproducibility, we ensured that all methods received identical training/validation/test splits (8:1:1 stratified).

**Evaluation metrics.** We evaluated model performance using five standard classification metrics: *Accuracy*, *Precision*, *Recall*, *F1-score*, and *Area under the ROC curve (AUC)*. Accuracy measures overall correctness, while precision and recall characterize positive prediction reliability and sensitivity, respectively. F1-score balances precision and recall, and AUC provides a threshold-independent measure of ranking quality. All metrics were computed on the held-out test set and reported as averages over three independent runs to mitigate variance.

**Training protocol.** All models, including baselines, were trained using the same stratified split and identical evaluation pipeline. For TopoMHC, we used AdamW optimizer with learning rate scheduling and early stopping; baseline methods retained their recommended default hyperparameters. This ensures that observed performance differences stem from modeling choices rather than inconsistencies in training or evaluation.

