# OpenReview forum: "TopoMHC: Sequence–Topology Fusion for MHC Binding"
_ICLR.cc/2026/Conference — ICLR 2026 Conference Withdrawn Submission_

### Official Review · Reviewer_wFSt · 2025-10-28

**Soundness:** 2
**Presentation:** 2
**Contribution:** 2
**Rating:** 4
**Confidence:** 3

**Summary:**

The paper introduces TopoMHC, a multi-modal framework that integrates protein sequence embeddings with topology-informed descriptors of peptide structures to predict immunogenicity. The framework combines ESM-based sequence representations with handcrafted geometric and persistent homology features, capturing both local sequence motifs and global structural organization. Experiments on a curated binding affinity dataset demonstrate that TopoMHC outperforms existing sequence-only and structure-based baselines. Ablation studies confirm the contributions of adaptive feature fusion and bidirectional cross-attention, highlighting the importance of integrating topological information for immunogenicity prediction.

**Strengths:**

1. The integration of sequence embeddings with topology descriptors through cross-attention is an innovative approach that enhances the expressiveness of the embeddings compared to sequence-only counterparts. Encoding structure information using topological descriptors is a meaningful attempt.
2. The paper demonstrates fairly competitive performance in peptide immunogenicity prediction, outperforming existing baselines.
3. The ablation study shows fairly definitive results on the effectiveness of individual components in the proposed method, namely the attentional feature fusion, cross attention, gating mechanism, and topology features.

**Weaknesses:**

1. The visual components of this paper are subpar. For example, Figure 1 and Figure 2 are not information dense enough as they waste a lot of white space.
2. The quantitative comparisons lack confidence intervals or similar statistical test results, which are especially necessary when the margin of improvement is not big.
3. The set of methods for comparison is relatively limited. I recommend discussing and optionally benchmarking against other recent methods for immunogenicity prediction, such as MHCflurry [1], DeepNeo [2], BigMHC [3] and ImmunoStruct [4].
4. This may be a little picky: based on my understanding, the authors are using the topology features as a representation of the peptide conformation. In that case, the authors may want to provide better rationales for not using the structure information directly (for example, using a graph neural network to encode the structure).

[1] MHCflurry 2.0: Improved Pan-Allele Prediction of MHC Class I-Presented Peptides by Incorporating Antigen Processing. Cell Systems, 2020.

[2] DeepNeo: a webserver for predicting immunogenic neoantigens. Nucleic acids research, 2023.

[3] Deep neural networks predict class I major histocompatibility complex epitope presentation and transfer learn neoepitope immunogenicity. Nature machine intelligence, 2023.

[4] ImmunoStruct: a multimodal neural network framework for immunogenicity prediction from peptide-MHC sequence, structure, and biochemical properties. bioRxiv, 2024.

**Questions:**

See Weaknesses.

---

### Official Review · Reviewer_1AF6 · 2025-11-01

**Soundness:** 3
**Presentation:** 3
**Contribution:** 2
**Rating:** 6
**Confidence:** 2

**Summary:**

This paper introduces TopoMHC, a novel multi-modal deep learning framework for predicting peptide-MHC binding, a critical task for vaccine design and immunotherapy. The key idea is to fuse information from two distinct modalities: sequence embeddings from a pre-trained protein language model (e.g., ESM-C) and topology-informed structural descriptors. To obtain the topological features, the authors first generate 3D conformers of the peptides using RDKit. From these conformations, they compute a suite of descriptors, including contact maps, interface geometry, distance statistics, and, most notably, persistent homology invariants. A sophisticated fusion mechanism is proposed to integrate these two modalities. It employs a bi-directional cross-attention module to align the token-level sequence embeddings with the topology features (represented as pseudo-tokens). The outputs are then combined using an Adaptive Feature Fusion (AFF) layer and a final gating mechanism. The authors evaluate TopoMHC on a curated dataset for the HLA-A*02:01 allele. The results demonstrate that TopoMHC achieves state-of-the-art performance, outperforming strong sequence-based baselines like NetMHCpan-4.0. An ablation study confirms that the fusion of topological features provides a measurable performance boost over a sequence-only (ESM-only) baseline

**Strengths:**

1. The primary strength is the innovative combination of a pre-trained PLM with transformation-invariant topological features (persistent homology). This provides a new and principled way to incorporate structural information that is robust to specific spatial coordinates.

2. The paper clearly details its multi-stage fusion mechanism (cross-attention, AFF, gating) and the specific topological features used (contact maps, interface geometry, PH) . This makes the work reproducible and the architecture easy to understand.

3. The ablation study (Table 2) provides clear evidence that the multi-modal fusion is effective. The performance of the full model is demonstrably better than the strong ESM-only baseline, confirming the utility of the topological features.

**Weaknesses:**

1.The model is only evaluated on a single MHC allele type. The MHC system is characterized by extreme polymorphism (thousands of alleles), and pan-allele generalization is the central challenge in the field. The paper's own introduction criticizes older models for "generalizing poorly across alleles", yet it provides no evidence that TopoMHC performs well on any allele besides the one it was trained on. This severely limits the scope and impact of the SOTA claim.

2. The topological feature pipeline adds substantial computational cost (generating 10 conformers per peptide, running UFF optimization, computing persistent homology). It is debatable whether the modest gain (over NetMHC)  justifies the significant increase in computational and methodological complexity for practical applications like large-scale screening.

3. There is a major contradiction in the paper's description of its inputs, which raises some questions about the methodology. This is detailed in "Questions" below.

**Questions:**

1. Is this model for peptide-MHC binding or antibody-antigen binding? If it is for peptide-MHC, why do the methods sections repeatedly refer to "antibody" and "antigen"? If $S_a$ is the peptide and $S_g$ is the MHC, the method for obtaining and using the MHC structure to calculate interface features is completely missing from the conformation generation section. Please clarify this fundamental point, as it is currently impossible to be certain what the model is actually doing.

2. Given the single-allele type evaluation, how do the authors expect the model to generalize? The topological features appear to be generated from the peptide alone (per Sec 3.3 ). How can such features help distinguish binding preferences for different MHC alleles, which have different binding groove structures?

---

### Official Review · Reviewer_5kkD · 2025-11-01

**Soundness:** 1
**Presentation:** 1
**Contribution:** 1
**Rating:** 0
**Confidence:** 4

**Summary:**

The paper tackles an important problem of prediction of peptide immunogenicity. Combining structural topological features and sequence features is a potentially interesting approach. However, there are several major issues with this paper (that I mention below) that lead me to suggest a low score.

**Strengths:**

The paper tackles an important problem of prediction of peptide immunogenicity. Combining structural topological features and sequence features is a potentially interesting approach. However, there are several major issues with this paper (that I mention below) that lead me to suggest a low score.

**Weaknesses:**

- Out of 33 references, I could not find the following 8 references (or 24%) after searching on Google, Google Scholar, Semantic Scholar, and checking their DOIs. In current times I can't help but imagine that these references may have been hallucinated by a LLM. I have therefore flagged this paper for potential significant LLM usage.

Here is the list of references I could not find:

H Cai et al. Gearbind: Geometry-aware binding affinity prediction for general macromolecule pairs. Nature Communications, 15:7785, 2024. doi: 10.1038/s41467-024-37785-9.

A Chakrabarty, R Singh, and S Kumar. Deepppapred: Deep learning for antibody–antigen binding affinity via patch-level interface learning. In Proceedings of the AAAI Conference on Artificial Intelligence. AAAI Press, 2025. Forthcoming / preprint available.

Gaurav K Ganotra and Rebecca C Wade. Predictive modeling of protein–ligand binding free energies using combine analysis. Journal of Chemical Theory and Computation, 14(1):170–183, 2018.

Xin Liu, Yuchen Fang, and Ling Zhao. Pareto-optimal persistent homology features for biomolecular binding prediction. Journal of Computational Chemistry, 43(12):820–834, 2022.

Yuchen Han, Yohan Kim, Dalibor Petrovic, Alessandro Sette, Morten Nielsen, and Bjoern Peters. Deepligand: a deep learning framework for peptide–mhc binding prediction. Bioinformatics, 39(1):btac834, 2023. doi: 10.1093/bioinformatics/btac834.

Claus Lundegaard, Ole Lund, and Morten Nielsen. Pickpocket: predicting binding specificities for receptors based on receptor pocket similarities. BMC Bioinformatics, 9(1):368, 2008b. doi: 10.1186/1471-2105-9-368.

Chi Y Pun, Marcio Gameiro, and Kelin Xia. Persistent homology of proteins: Topological data analysis of protein flexibility, dynamics, and folding. Bulletin of Mathematical Biology, 80(7):2005–2033, 2018. doi: 10.1007/s11538-018-0429-6

Lee-Ping Wang, Dominic Wu, Chi-Tak Wong, Troy van Voorhis, Vijay S. Pande, Daniel J. Cole, Ethan C. John, David L. Mobley, Pavel V. Klimovich, Caitlin C. Bannan, Andrea Rizzi, and John D. Chodera. Accurate and reliable prediction of relative ligand binding potency in prospective drug discovery by way of a modern free-energy calculation protocol and force field. Journal of Chemical Theory and Computation, 15(5):3510–3524, 2019. doi: 10.1021/acs.jctc.9b00287

- In addition, the following references have wrong bibliographic information:

Birkir Reynisson, Bjoern Alvarez, Sinu Paul, Bjoern Peters, and Morten Nielsen. Netmhcpan-4.2: improved prediction of cd8+ epitopes by use of transfer learning and structural features. Frontiers in Immunology, 16:1616113, 2025. doi: 10.3389/fimmu.2025.1616113

Correct citation is:

Nilsson, J. B., Greenbaum, J., Peters, B., & Nielsen, M. (2025). NetMHCpan-4.2: improved prediction of CD8+ epitopes by use of transfer learning and structural features. Frontiers in Immunology, 16, 1616113.


Raphael J L Townshend, Stephan Eismann, Andrew M Watkins, Rishab Rangan, Maria Karelina, Rajesh Das, John Jumper, Gerard J Kleywegt, Ashesh Krishnapriyan, and Ron O Dror. Atom3d:Tasks on molecules in three dimensions. In NeurIPS, volume 34, pp. 14910–14924, 2021

Correct citation at: https://openreview.net/forum?id=FkDZLpK1Ml2


- Figure 1, explaining the central workflow of the proposed method, is very poorly described. See Questions below.

**Questions:**

- Please explain what is RDKit, which is mentioned several times in the paper but is never defined. A reference to original papers implementing this method would also help.

- Figure 1 is very confusing and the caption does not explain much. The first step of the sequence processing uses a "pre-trained protein language model" such as ESM-C to embed the sequences. What is a "Batch Converter" ? What is "Sequence Clippin" ?  Why is there a subsequent "ESM inputs Embedding" step, if the ESM was already used at the first step ?

- Again, on Figure 1, for the topological processing, what is "concatenation" ? Why is "Persistent Homology" depicted as part of "conformation generation" ?

- Section 3.2 opens with "We concatenate the antibody and antigen sequences" ... why is there an antibody involved here? I thought the paper was talking about MHC presentation of peptides (i.e., immunogenicity), which are eventually recognized by a TCR, not an antibody. However, focusing on immunogenicity only, not even the TCR sequence needs to be mentioned here !

- Section 3.3 says "we construct 3D conformations using RDKit with "ETKDG initialization and UFF optimization". What is ETKDG, UFF ? Please define these terms, citing literature of needed.

- Also in Section 3.3, it says "Interface Geometry Features. For two chains (antibody and antigen), [...]" . Again, why is there an antibody mentioned here?

- Please number all equations in the paper.

**Details Of Ethics Concerns:**

The paper seems AI generated, and several central points simply made no sense to me after reading it.

---

### Official Review · Reviewer_1J9q · 2025-11-01

**Soundness:** 4
**Presentation:** 4
**Contribution:** 4
**Rating:** 8
**Confidence:** 4

**Summary:**

This paper introduces TopoMHC, a multi-modal deep learning framework designed to predict peptide-MHC binding immunogenicity. The core innovation of this paper is the combination of sequence-based embeddings from a pretrained PLM with topology-informed structural descriptors derived from 3D peptide conformations. The sequence and topological features are integrated using a bi-directional cross-attention mechanism and an adaptive fusion layer. The authors demonstrate that TopoMHC achieves state-of-the-art performance on a curated dataset for the HLA-A*02:01 allele, significantly outperforming sequence-only baselines in recall.

**Strengths:**

- This paper tackles an important problem in drug discovery and shows improved performance over netMHCpan, which is currently widely adopted as peptide-MHC binding oracle
- The primary strength is the integration of topological data analysis, specifically persistent homology, into the prediction pipeline, capturing structural invariants robust to noise and spatial transformations.
- The model architecture is well thought out and employs a nice fusion strategy to weight different contributions.

**Weaknesses:**

- Evaluation on a single MHC allele, so that generalizability for the stated goal of vaccine design and immunotherapy improvements is unclear as that requires pan-allele coverage.
- The model achieves modest improvement over the NetMHCpan-4.2 baseline and the ESM-only ablation, but this comes at a substantial computational price of generating multiple conformers per peptide and RDKit and UFF optimization, then computing complex topological features like persistent homology for each.

**Questions:**

- Could the authors provide a summary of inference compute needed for each method in their benchmark

---

### Note · Authors · 2025-12-01

I have read and agree with the venue's withdrawal policy on behalf of myself and my co-authors.